# Local Myoelectric Sensing During Human Colonic Tissue Perfusion

**DOI:** 10.3390/diagnostics14242870

**Published:** 2024-12-20

**Authors:** Matan Ben-David, Raj Makwana, Tal Yered, Gareth J. Sanger, Charles H. Knowles, Nir Wasserberg, Erez Shor

**Affiliations:** 1Townsville Hospital and Health Service, Douglas, QLD 4814, Australia; 2Exero Medical Ltd., Or Yehuda 6037606, Israelerez.shor@exeromedical.com (E.S.); 3College of Medicine and Dentistry, James Cook University, Townsville, QLD 4814, Australia; 4Blizard Institute, Faculty of Medicine and Dentistry, Queen Mary University of London, London E1 4NS, UK; r.makwana@qmul.ac.uk (R.M.); c.h.knowles@qmul.ac.uk (C.H.K.); 5Rabin Medical Center, Faculty of Health and Medical Sciences, Tel Aviv University, Tel Aviv 6997801, Israel; nirw@clalit.org.il

**Keywords:** anastomotic leakage, human colon

## Abstract

**Objectives:** Anastomotic leakage (AL) is one of the most devastating complications after colorectal surgery. The verification of the adequate perfusion of the anastomosis is essential to ensuring anastomosis integrity following colonic resections. This study aimed to evaluate the efficacy of measuring the electrical activity of the colonic muscularis externa at an anastomosis site for perfusion analysis following colorectal surgery. **Methods:** Strips of human isolated colon were maintained in a horizontal tissue bath to record spontaneous contractions and myoelectric activity and spike potentials (using a bipolar electrode array for the wireless transmission of myoelectric data—the xBar system) from the circular muscle. Intraoperative myoelectric signal assessment was performed by placing the electrode array on the colon prior to and following mesenteric artery ligation, just prior to colonic resection. **Results:** In human isolated colon, the amplitude, duration, and frequency of contractions were inhibited during hypoxia by >80% for each measurement, compared to control values and time-matched oxygenated muscle. Intraoperative (*N* = 5; mean age, 64.8 years; range, 54–74 years; 60% females) myoelectric signal assessment revealed a decline in spike rate following arterial ligation, with a mean reduction of 112.64 to 51.13 spikes/min (*p* < 0.0008). No adverse events were observed during the study, and the device did not substantially alter the surgical procedure. **Conclusions:** The electrical and contraction force of the human colon was reduced by ischemia, both in vitro and in vivo. These preliminary findings also suggest the potential of the xBar system to measure such changes during intraoperative and possibly postoperative periods to predict the risk of anastomotic viability as a surrogate of evolving dehiscence.

## 1. Introduction

Anastomotic leakage (AL) is one of the most devastating complications after colorectal surgery [1]. The clinical consequences of AL include increased perioperative mortality and morbidity, along with a significant impact on health-related quality of life and overall economic healthcare burden [2,3]. The incidence of AL in elective cases can vary between 1 and 19% depending on multiple clinical and perioperative factors, such as the anatomic location of the anastomosis, the presence of associated high-risk comorbidities, and the use of neoadjuvant chemoradiation or postoperative adjuvant chemoimmunotherapy [1]. In addition, a leading contributor is the poor perfusion of the anastomosis [4,5]. Surgical decision making concerning the optimal site for an anastomosis is traditionally based on subjective intraoperative indicators that define bowel viability. These include the clinical assessment of the color of the intestinal wall, the observation of peristalsis, active bleeding from the edges of the resection, and visible or palpable mesenteric arterial pulsation in the anastomotic vicinity [6].

It is appreciated, however, that the surgeon-predictive capacity for AL is inconsistent and potentially low [7], with many centers now introducing intraoperative fluorescence angiography for a more objective real-time determination of the perfusion of the intestinal ends, before and after anastomosis. In several meta-analyses of non-randomized studies, this approach, using a variety of fluorophores, has shown consistent reductions in the incidence of AL [8,9,10,11] and has altered the surgical decision making in about 10% of cases where it was employed [12]. While encouraging, further improvements in successful prediction are needed.

An alternative approach to predicting AL is to measure the electrical activity of the colonic *muscularis externa* at the anastomosis, using a methodology capable of detecting changes in cell-based physiology in real time. Historically, measurements of the electrical activity of the gastrointestinal (GI) tract in patients have focused on the diagnosis of a variety of upper GI disorders, using indirect electrogastrographic techniques [13]. This involves the use of electrodes placed on the surface of the skin to detect the sum of slow-wave electrical potentials generated within the stomach wall. These are initiated and propagated by syncytial networks of pacemaker-type cells (the interstitial cells of Cajal (ICC) together with platelet-derived growth factor receptor-α + cells) that electrically couple with smooth muscle cells. The slow waves, in turn, can generate phasic muscle contractions and also facilitate nerve-muscle signaling [14]). More recently, slow-wave electrical activity has been recorded in patients via multiple electrodes placed directly on the stomach serosa [15]. However, such techniques have not been widely used in patients with lower GI disorders [13,16,17].

We recently established small and large animal in vivo models of AL, in which an electrode array was placed on the serosal surface of the colon [18]. This bespoke system (xBar^TM^) has been further developed to detect AL in human surgical patients [19], collecting bioimpedance variations caused by hypoxia or local inflammation for wireless relay and analysis. In the present study, we have used the same system, with two aims. Firstly, to study the effects of temporary hypoxia on the functions of human isolated colon, measuring surface electromyography, and secondly, muscle contraction force. The latter was included to determine if changes in myoelectrical activity reflected changes in muscle functions. The second aim was to compare the results obtained in vitro with intraoperative electromyography recordings made during colorectal resection before and after the ligation of the main arterial pedicle. Overall, the results suggest that, in humans, hypoxia and its consequences on muscle contractility can be reliably sensed by measurements of myoelectric activity, potentially providing early detections of AL for early intervention to reduce the devastating sequela of leaks.

## 2. Materials and Methods

### 2.1. Patients

All studies were approved by local hospital ethics committees at the participating centers: ex vivo recordings on colonic tissue (UK REC 15/LO/2127); intra-operative recordings (Rabin Medical Center, Petach-Tikva, Israel RMC 19-0863). All patients, in both studies, provided written informed consent.

#### Myoelectric Signal Acquisition and Analysis

The xBar system (Exero Medical, Or Yehuda, Israel) was used to acquire myoelectric activity in all experiments. In brief, the xBar system is a bipolar electrode array connected to a microprocessor-controlled A2D converter for biopotential measurements (Texas Instruments ADS1299, Dallas, TX, USA) [18,19]. The xBar is capable of recording, storing, and wirelessly transmitting myoelectric data. For intra-operative recordings, the electrode array was embedded in a standard surgical Jackson–Pratt drain used as a conduit to enable clinical localization of the electrode array near the surgical site. In its final intended clinical use, the xBar monitoring system collects myoelectric data (local field potential recorded continuously from several points at typical distances of 5–400 mm from the surgical site), transmits these data for processing, and may provide clinical warnings if data patterns are abnormal or indicate undesired physiological conditions (Figure 1).

### 2.2. Human Isolated Large Bowel

Macroscopically normal full-thickness specimens of human bowel were obtained from 4 female patients (median age 61, range 48–80 years). Specimens were cut 5–10 cm from the tumor from the descending colon (*n* = 2), sigmoid colon, and rectum (*n* = 1 each). Tissues were immersed in pre-oxygenated Krebs solution (mmol/L: NaCl, 118.3; CaCl_2_, 2.5; KH_2_PO_4_, 1.2; KCl, 4.7; MgSO_4_, 1.2; NaHCO_3_, 25; glucose, 11.1, equilibrated with 5% CO_2_/95% O_2_) within 40 min after surgery and handled in accordance with the principles articulated for neuropharmacological assessment [20]. Upon arrival at the laboratory, the mucosa, submucosa, and muscularis mucosa were removed by a combination of sharp and blunt dissection and discarded. Up to two muscle strips (10 × 30 mm) were used from each colon specimen, cutting approximately parallel to the circular muscle fibers from the regions between the *taenia coli.* These were stored overnight (15–18 h) in fresh oxygenated Krebs solution at 4 °C for use the following day. Previous experiments have shown that this procedure does not significantly affect spontaneous contractions or electrical-field-stimulation-evoked cholinergic contractions and/or nitrergic relaxations of the muscle [21,22]. During use, muscle strips were placed in a horizontal 100 mL bath containing Krebs–Henseleit solution that was pre-gassed with 95% O_2_/5% CO_2_, maintained at 37 °C, and renewed at 5 mL/min. Two-thirds of each strip was pinned down at its edges onto a silicone (Sylgard 184, Dow Corning, Barry, UK) layer that lined the bottom of the bath. This area was used for myoelectrical recording from the serosal surface using the xBar recording electrode. The ground electrode was placed in the bath solution surrounding the tissue. The opposite, free end of a muscle strip was attached to a pre-calibrated MLT201/D force transducer (AD Instruments, Chalgrove, UK) using a cotton suture for measuring isometric contractions and an AcqKnowledge version 3.8.1 data acquisition system (BIOPAC Systems Inc., Goleta, CA, USA).

The strips were stretched by 20 mN and allowed to equilibrate for 150 min. Spontaneous contractions of the muscle strips were recorded, together with recordings of myoelectric activity. From each specimen, one strip was prepared as a control (time-matched and vehicle) and another for hypoxia testing. Hypoxic conditions were established by disconnecting the reservoir containing Krebs solution (gassed with 95% O_2_/CO_2_), stopping the bubbling of 95% O_2_/CO_2_ into the solution bathing the tissue, and then circulating fresh un-gassed Krebs solution at the same flow rate (Figure 2). The results from each tissue were merged (*n* = 4 total) for analysis and radar plotting of the different parameters of electrical activity and muscle movements (see [22] for method of analysis).

The contraction force and the spiking activity observed in the myoelectric data were tightly correlated, and the contractions were also mathematically reconstructed from the myoelectric data (Appendix A).

### 2.3. Intraoperative Myoelectric Signal Assessment After Mesenteric Artery Ligation

The xBar system was used on patients undergoing low anterior resection surgeries. During surgery, baseline myoelectric signals were recorded with a Smart Drain for an initial 4 min period by placing the electrode array on the colon 1 cm distal to the projected site of the proximal specimen anastomosis line. Without changing the course of the surgery, myoelectric data were then re-measured after inferior mesenteric artery ligation just prior to colonic resection (time interval recorded).

### 2.4. Data Analysis and Statistics

All experiments were performed in parallel with relevant vehicle-treated and time-matched controls. Further, myoelectric activity recordings were synchronized with basal muscle tone recordings. Recorded spontaneous contractions in the absence (or presence) of an agonist were quantified using a custom code written in MATLAB R2020a (The Mathworks, Natick, MA, USA). The parameters measured included the amplitude of spontaneous contractions (gram tension, g), the area under the contraction wave (g.s), the duration of the contraction (s), and the interval between contractions (s).

Myoelectric data were analyzed using MATLAB (The Mathworks, MA, USA). Signals were high-pass-filtered (2 Hz cutoff, Butterworth 5th order) and subsequently threshold-filtered to obtain individual spikes. To reconstruct the contraction, under the assumption that the contraction force can be modeled by a linear-time-invariant system derived from the spiking activity, we calculated the convolution of the filtered spike train and the hypothesized transfer function of the tissue, which was taken as a rectangular window. As contractions represent the cumulative effect of multiple action potentials, a good correlation was noted between the recorded spikes and mechanical activity (Appendix A). In view of these findings, the waveforms were analyzed by reconstructing the contraction patterns from the spike-wave activity (and vice versa) to establish parameters of interest for the preliminary in vivo study with the Smart Drain. Waveform analysis used the mathematical approach of two coupled oscillators, as described by Schwartz, Harvard MA [23]. For this analysis, mechanical contraction curves were recreated using only the myoelectric data by filtering out the spike potential thresholds, resulting in a good closeness of fit between the reconstructed and measured mechanical wave patterns. The reverse approach was then applied to convert a contraction into a spike potential pattern by fitting the mechanical section between the estimated spike train and the window function.

Where appropriate, a paired or unpaired Student’s *t*-test was performed for comparisons of individual means of data from within a given specimen or between specimens from different donors, respectively, with a *p*-value < 0.05 considered statistically significant.

## 3. Results

### 3.1. The Effects of Hypoxia on Myoelectric Signals and Muscle Contractility In Vitro

Myoelectric signals and muscle contractions were recorded under normal (oxygenated) and hypoxic conditions, as illustrated in Figure 2A. These were similar, but not identical, between the different tissues used (as indicated by the error bars in the radar plot; Figure 2C). Within a given tissue, there was a tendency for the contraction waveform to vary, with some being monophasic and others biphasic at times. However, irrespective of tissue type, under control conditions, a burst of relatively high-amplitude myoelectric activity coincided with distinct, phasic muscle contractions, which occurred spontaneously and rhythmically; the contractions mostly exhibited a monophasic waveform of rapid onset but slow recovery (Figure 2B). The contraction amplitude (1.6 ± 0.3 g), duration (124.5 ± 16.9 s), area under each contraction wave (29.2 ± 15.0 g.s), and interval between contractions (90.3 ± 12.6 s) of all four specimens are presented as a radar plot to show synchronization with the spike electrical activity of the muscle (Figure 2C). Under hypoxic conditions, the amplitude, duration, and frequency of contractions were inhibited by more than 80% for each measurement (Figure 2C) compared to control values and time-matched oxygenated muscle strips.

### 3.2. Spiking Activity Declines in Ischemic Conditions In Vivo

Five patients (mean age, 64.8 years; range, 54–74 years; three female) underwent open (*n* = 1), robotic (*n* = 1), or laparoscopic (*n* = 3) anterior resection of rectal cancer over a 3-month period. Anastomosis was performed between the descending colon and rectum following visual confirmation of vascularity but without the use of intra-operative ICG.

Baseline myoelectric activity was recorded by the xBar system for a minimum of 4 min. In the laparoscopic cases, the xBar recording was repeated approximately 60 min after de-vascularization and in the open case, 20 min after the ligation of the main segmental colonic blood supply. Similarly to the in vitro results, myoelectric activity typically presented spike bursts followed by periods of pausing. Burst–pause cycles were observed in a typical frequency of 2–4 cycles/minute. A decline in the spike rate from baseline was noted overall (Figure 3A), qualitatively similar to that observed during the ex vivo studies. Quantitatively, the decline in the spiking rate was statistically significant (pre-ligation = 112.64 ± 16.13 spikes/min vs. post-ligation = 51.13 ± 24.88 spikes/min; *p* < 0.0008, Figure 3B). No adverse effects were noted in any patient with the placement of the xBar system.

## 4. Discussion

This study has demonstrated the ability of the xBar system to detect myoelectric activity in the human colon, in vitro and in situ, prior to colonic resection. In the isolated tissues, synchrony was also observed between a period of relatively high-amplitude electrical activity and the occurrence of muscle contractions. The mechanisms were not investigated, but others suggest that synchrony can occur in the human colon and stomach following the conduction of slow waves generated by ICCs into the smooth muscle cells [24,25]. In other experiments with human isolated colon, synchrony between the myoelectric activity of the circular muscle (recorded in mucosa-free preparations by suction electrodes on the muscle) and muscle contraction has been observed only after pacing using carbachol added to the bathing solution [26]. Differences in methodology likely explain the different observations. In addition, for the present experiments conducted in vitro, small variations in rhythmicity between tissues from different patients may have been caused by differences in the dominance of different populations of ICCs between the tissues, now removed from extrinsic nerve control. Further experiments with a greater number of tissues are required to confirm this variation.

The assessment of myoelectrical activity in the human colon is not new [13,27], with most studies focusing on the basic rhythmicity and propagation of variable-frequency-pattern slow waves, representing changes in membrane potentials generated predominantly by the ICCs [28], a pacemaker cell syncytium, which can be considered to act like a weakly coupled oscillator. Thus, different populations of ICCs may initiate, propagate, regenerate, and maintain electrical activity (frequency and coupling strength) over relatively long distances [28,29,30]. Together with enteric or extrinsic nerve activity, a threshold is reached during which rapid depolarization or spike activity appears, overlaying the background slow-wave plateau potentials to generate (for example) propulsive and retropulsive movements in the colon [29]. In mouse colon, serosal electrodes have recorded neurogenically controlled spike bursts and myogenic spike bursts, which are pacemaker-generated [31]. Similar patterns of cyclical motor complexes have been shown in whole human isolated colon (without extrinsic innervation) [32], although with lower frequencies than with in vivo manometric recordings [33]. Recently, Lin et al. [34] measured the spike activity and cyclic motor patterns prior to and following distal colonic resection, using a multielectrode array that was laparoscopically applied to the colonic serosa. However, these authors were unable to consistently demonstrate a propagating electrical wave. By contrast, spike waves have been recorded extracellularly in the dog colon and rectum, tending to terminate after very short propagation distances [35,36].

There has been a recent surgical emphasis directed toward more objective intraoperative measures that assess bowel viability prior to anastomosis and which are specifically aimed at the prevention of AL. Several systematic reviews and meta-analyses have evaluated the impact of Indocyanine Green (ICG) fluorescence imaging [8,9,10,11,12] over the more usual subjective examination, which includes visual inspection under white light, determination of color and peristalsis, the observation of visible pulsation in the marginal artery, and the presence of active bleeding from the resected edges of the bowel [37]. The intraoperative use of Doppler-detected intramural blood flow has not shown any advantage in the diagnosis of on-table ischemia [38]. However, the use of infrared imaging and quantitative software analyses of these fluorophore perfusion signals [11] has been shown to detect regions of low or absent fluorescence and reduce the risks of AL through an improved estimate of bowel microperfusion [39]. Notably, the relative success of such adjuncts, specifically aimed at anastomotic perfusion, is affected by technical factors such as the dose of ICG infused, as well as patient-related considerations concerning higher-risk anastomoses [40].

A range of ischemia–reperfusion animal models have been developed [41] that show changes in electrical parameters with ischemia in the GI tract and other tissues using electrical impedance spectroscopy [42]. The present experiments now demonstrate the utility of the xBar system in detecting ischemia-induced changes in myoelectrical activity in vitro and, again, in vivo, following mesenteric arterial ligation, supporting its use in detecting the development of AL following surgery [19]. These experiments can both be impacted by the degree and duration of ischemia. In the earlier phases, there is a decrease in the amount of extracellular water as cellular ionic pumps fail, and the cells swell with more prolonged ischemia, leading to altered cell membrane permeability [43,44]. By contrast, using a unique spike detection algorithm, Erickson et al. [45] showed that mesenteric ligation in the small bowel resulted in an increase in spike activity, which then decreased 20 min after the induction of 100% ischemia. This multielectrode array assessed the contractile propagation and demonstrated conduction blockade with self-terminating, anisotropic circumferential, and retrograde propagating wavefronts. Similarly, in cat small intestine [46], acute ischemia caused a rapid breakdown in the organized propagation of slow waves (depression of conduction, local in-excitability, conduction block, and the appearance of subsidiary pacemakers), changes that find some consistency with the present findings. For example, although the experiments with human isolated colon did not measure the propagation of myogenic activity, the introduction of ischemia caused a clear inhibition of the frequency of rhythmic contractions together with the amplitude and duration of the contractions.

Our study has several limitations. Firstly, the recordings taken from segments of human isolated colon showed spontaneous myoelectrical properties that differed from those measured in vivo because of the disruption of the extrinsic and enteric neural pathways that help to control ICC activity and neurogenic motor patterns [26]. Nevertheless, the model does enable the xBar system to be evaluated in human tissue and for correlations between electrical and mechanical activity to be examined. Further, in the present experiments with human isolated colon, ischemia reduced both electrical and mechanical activity to a similar degree, regardless of the small variations in the patterns of activity observed between the different tissues prior to ischemia. Additional limitations concern the technique and accuracy of serosal myoelectric GI mapping, where it is appreciated that there is no clearly defined relationship between electrophysiological events and observable motor activity [47]. In this respect, recordings can be influenced by the type of electrode assembly (multiarray vs. close-spaced bipolar or monopolar), as well as by the hardware and software used for data acquisition and signal filtering [29,47]. Although bipolar recordings have a higher signal-to-noise ratio (SNR), they are less sensitive in the detection of repolarization and can be affected by how the electrodes are oriented to a propagating wavefront [48]. In this study, we attempted to collect local spiking information driving the colonic smooth muscle. Using a small, close-spaced multiarray, we were able to record data from several vectors to obtain components that are aligned to the longitudinal muscles and perpendicular to them (collecting spiking information from the circular muscles).

Slow-wave recording using an electrode array is more challenging. Serosal mapping in the colon may not adequately detect deeper ICC activity [49] and is also affected by the relatively low SNR, as well as by the colonic geometry [50]. Only some of these effects are obviated by the use of a simple single recording electrode designed to assess basal myoelectric functions. Differences in the mathematical modeling that analyzes slow waves, or that detects spike thresholds, also make strict comparisons between studies difficult, particularly as typical spike potentials share considerable spectral overlap with the faster downward deflections that are normally part of recognizable slow waves [51].

A further limitation is that the effects of general anesthesia and colonic manipulation on spike activity add to the complexity of interpreting intraoperative myoelectrical activity where a system such as the xBar would be used for the early detection of anastomotic ischemia [32]. If its use is to be expanded as a postoperative early detection warning alarm for AL, future work will need to determine the colonic myoelectrical patterns of uncomplicated resection and anastomosis [52,53] and factor in the effects of postoperative opioid analgesia on normal spike patterns [54]. Studies assessing bioimpedance during ischemia–reperfusion cycles in an in vivo animal model have shown high reliability using a learning feed-forward neural network algorithm for the binary prediction of tissue viability from a single measurement [55].

## 5. Conclusions

Our in vitro and in vivo experiments suggest that xBar^TM^ recordings of electrical activity in the human colon are affected by ischemia. There is, therefore, the potential for the intraoperative and possibly postoperative use of this detection system to predict the risk of anastomotic viability as a surrogate of evolving dehiscence. Further work with laparoscopically or robotically positioned electrodes is required to establish the role of recordings made near uncomplicated anastomoses.

## Figures and Tables

**Figure 1 diagnostics-14-02870-f001:**
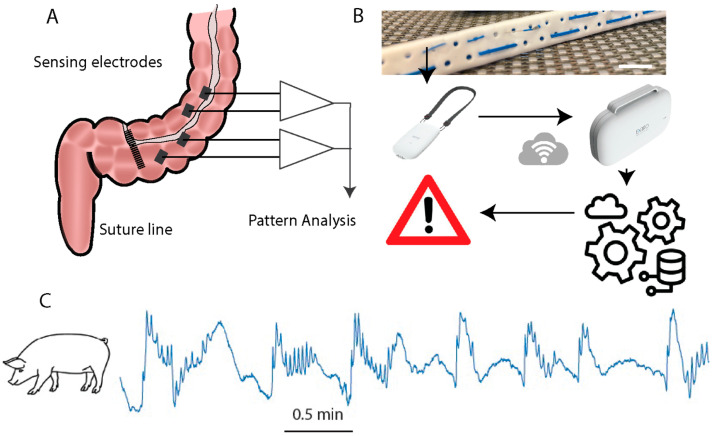
xBar myoelectric recording experimental system. (**A**) Conceptual myoelectric activity recording from near and farther from the anastomosis–suture line following colonic surgeries. (**B**) Electrode array is embedded in a standard surgical drain used to place the electrodes without changes in the surgical workflow. Data are recorded by a wearable device and then relayed wirelessly to the cloud where they are processed, alerting the clinical team if needed. (**C**) Typical representative signals collected previously by xBar from a colon in a pig model (recordings not previously shown)

**Figure 2 diagnostics-14-02870-f002:**
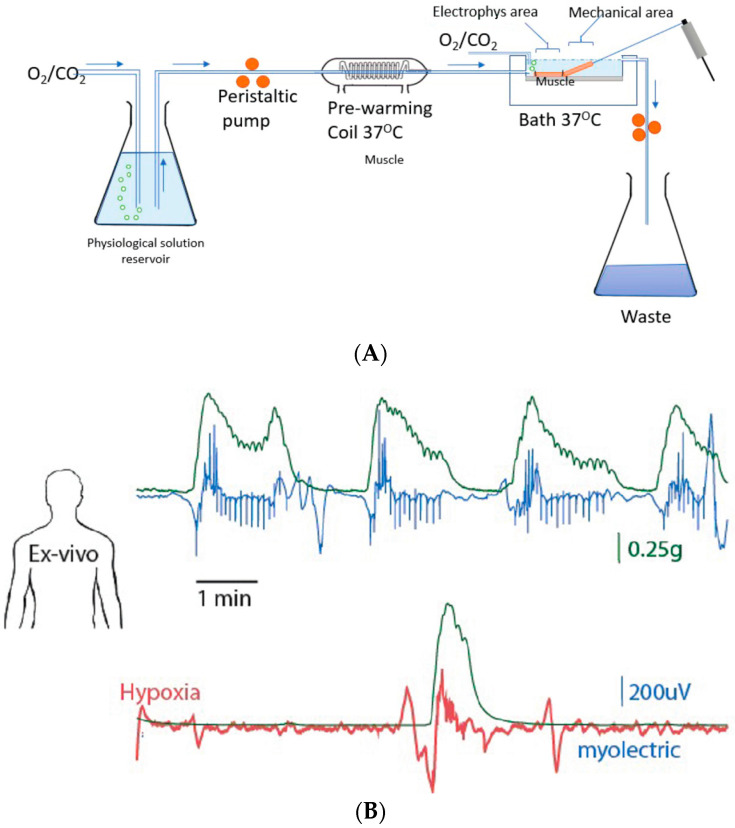
(**A**) Schematic diagram of the ex vivo setup used to collect contraction force and myoelectric activity simultaneously during normal and hypoxic conditions. (**B**) Representative contraction force signals (green), normal myoelectric signals (blue), and myoelectric signals in hypoxic conditions (red). (**C**) Radar plot illustrating spontaneous contraction features and myoelectric spike activity during normal and hypoxic conditions. Data are mean ± SEM, *N* = 3 patients, and each data set is significantly different (*p* < 0.05, paired *t*-test).

**Figure 3 diagnostics-14-02870-f003:**
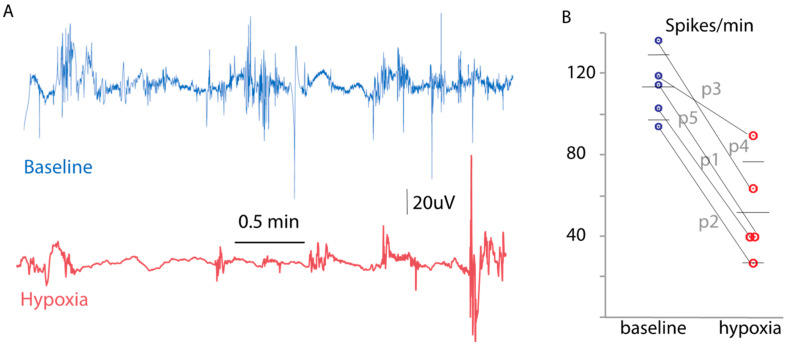
Myoelectric activity in normal and ischemic conditions in humans. (**A**) Representative baseline myoelectric activity (blue) and activity in ischemic colon following IMA devascularization (red). (**B**) Spiking activity before and after devascularization (*N* = 5, *p* < 0.0008).

## Data Availability

The original contributions presented in the study are included in the article/Appendix A, further inquiries can be directed to the corresponding author.

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
