# Peer review of "Local Myoelectric Sensing During Human Colonic Tissue Perfusion"

_diagnostics, 2024, doi:10.3390/diagnostics14242870_

Round 1
Reviewer 1 Report
Comments and Suggestions for Authors
The introduction describes the subject studied and presents clearly the aims of the article.
The materials and methods section presents in detail the equipment used for the study and the design of the study with multiple figures illustrating the experiment itself.
The results shows clearly, in detail, all the parameters obtained in the study, and the discussion section compares the method used in this study with other methods used for assessing the viability of the anastomoses from other studies.
The conclusions are supported by the results and are well stated at the end of the article.
Reviewer 2 Report
Comments and Suggestions for Authors
The manuscript provides a novel approach for detecting anastomotic leakage (AL) using the xBar system to monitor myoelectric signals of the colon. While the methodology is robust, some sections require minor refinements for clarity and completeness. The introduction should better emphasize the clinical implications of the findings, particularly highlighting the potential for the xBar system to serve as a standard intraoperative tool. Additionally, the methods section could benefit from more details on patient selection criteria for tissue acquisition and handling of outliers or missing data during analysis. Enhancing these areas will improve the transparency and reproducibility of the study.
The results and discussion sections are well-structured, but they could further emphasize key findings and their implications. The in vitro radar plot (Figure 2C) should be better integrated into the narrative, with explicit discussion of parameter trends under hypoxic conditions. Similarly, the in vivo findings could include additional demographic details, such as differences in surgical approaches, to contextualize the results. The discussion should expand on the potential for postoperative applications of the xBar system and acknowledge the need for external validation through larger-scale, multi-center studies. Minor technical revisions, including improved figure annotations, clearer labeling, and consistent terminology, will enhance readability and data presentation.
